# A realistic mixture of ubiquitous persistent organic pollutants affects bone and cartilage development in zebrafish by interaction with nuclear receptor signaling

**Gustavo Guerrero-Limón**[1], **Jérémie Zappia**[2], **Marc Muller**[1]*

**1** Laboratory for Organogenesis and Regeneration, GIGA Institute, University of Liège, Liège, Belgium,
**2** Bone and Cartilage Research Unit, Arthropôle Liège, Center for Interdisciplinary Research on Medicines (CIRM) Liège, Institute of Pathology, CHU-Sart Tilman, University of Liège, Liège, Belgium

* m.muller@uliege.be

## Abstract

"Persistent organic pollutants (POPs)" have a plethora of deleterious effects on humans and the environment due to their bioaccumulative, persistent, and mimicking properties. Individually, each of these chemicals has been tested and its effects measured, however they are rather found as parts of complex mixtures of which we do not fully grasp the extent of their potential consequences. Here we studied the effects of realistic, environmentally relevant mixtures of 29 POPs on cartilage and bone development using zebrafish as a model species. We observed developmental issues in cartilage, in the form of diverse malformations such as micrognathia, reduced size of the Meckel's and other structures. Also, mineralized bone formation was disrupted, hence impacting the overall development of the larvae at later life stages. Assessment of the transcriptome revealed disruption of nuclear receptor pathways, such as androgen, vitamin D, and retinoic acid, that may explain the mechanisms of action of the compounds within the tested mixtures. In addition, clustering of the compounds using their chemical signatures revealed structural similarities with the model chemicals vitamin D and retinoic acid that can explain the effects and/or enhancing the phenotypes we witnessed. Further mechanistic studies will be required to fully understand this kind of molecular interactions and their repercussions in organisms. Our results contribute to the already existing catalogue of deleterious effects caused by exposure to POPs and help to understand the potential consequences in at risk populations.

## Introduction

Persistent Organic Pollutants (POPs) are toxic chemicals that possess characteristics of special concern, they do not degrade easily, hence persisting in the environment for long periods of time. In addition, they bioaccumulate and are transferred through the food chain, exerting their effects at many different levels within the environment and potentially also in humans. POPs are particularly concerning for countries within the European Union where the ageing

**Data Availability Statement:** RNA-Seq data are accessible through GEO Series accession number

GSE208019 (https://www.ncbi.nlm.nih.gov/geo/query/acc.cgi?acc=GSE208019).

**Funding:** This research was funded by the European Union's Horizon 2020 research and innovation program under the Marie Skłodowska-Curie Innovative Training Network (ITN) program PROTECTED [Grant agreement No. 722634]. G.G-L. was a PROTECTED fellow, M.M. is a "Maître de Recherche" at "Fonds National de Recherche Scientifique (FNRS).

**Competing interests:** The authors have declared that no competing interests exist.

**Abbreviations:** ACT, Angle between ceratohyals; Br, Brominated compounds; Br+Cl, Dual mixture of brominated and chlorinated compounds; Cl, Chlorinated compounds; DCH, Distance between the frontal end of the ceratohyals and the line connecting the posterior ends of the hyosymplectics; LC, Length of ceratohyals; MPH, Distance covering the entire Meckel's cartilage, palatoquadrate, and the hyosymplectics; MPQ, Distance between the left and right Meckel's cartilage/Palatoquadrates; PFAA, Perfluorinated compounds; PFAA+Br, Dual mixture of perfluorinated and brominated compounds; PFAA+Cl, Dual mixture of perfluorinated and chlorinated compounds; POP125×, Total mixture of 29 perfluorinated, brominated, and chlorinated compounds at 125 times the average concentration found in blood.

population is high and there is a high degree of industrial activity that still relies on the use of POPs for daily life products such as plasticizers, flame retardants in electronic devices, furniture and fire-resistant clothing, and even in our kitchens with the non-stick cookware [1, 2]. Hence, regulations, survey and especially epidemiological studies must be paramount in places where ageing populations are large and increasing [3].

Several studies have found links between exposure to POPs and a plethora of adverse effects [4–10]. Moreover, being widely spread in the environment and some of our food sources (*e.g.*, seafood) [11, 12], POPs are under constant surveillance, however monitoring programs mostly consider each compound individually. A more holistic approach is needed as POPs are rarely found completely alone [13–16]. Investigations of POPs as mixtures present their own set of challenges and their biological and epidemiological implications are complex to understand. Increasingly, researchers in the past couple of years have addressed the problem of POPs as mixtures by investigating their effects using *in vitro*, *in vivo* and *in silico* approaches. Several studies have used constructed mixtures based on levels actually found in the blood of a Scandinavian human population [17]. This particular mixture is made of 29 compounds found at high levels in food, blood, and breast milk. Polychlorinated dibenzodioxins/polychlorinated dibenzofurans (PCDD/PCDF) and dioxin-like polychlorinated biphenyls (PCBs) were thus deliberately excluded. Relative concentrations of the compounds were based on estimated daily intake levels from Scandinavian studies. In addition, several sub-mixtures were designed, containing either chlorinated (Cl), brominated (Br), or perfluorinated compounds (PFAA) to be able to assign specific effects to one of these classes [17]. The total POP mixture was shown to antagonize the androgen receptor transactivation and nuclear translocation [18], inhibit the transactivation activity of the aryl hydrocarbon receptor [19], and to enhance the nerve-growth-factor-induced neurite outgrowth in PC12 cells at high concentration [18–21]. It also induces cytotoxicity and some of the sub-mixtures affect the number of cells, nuclear area and mitochondrial membrane potential in human A-498 kidney cells [21]. Recently, zebrafish larvae exposed to this POP mix at realistic concentrations, or sub-mixtures thereof, presented growth retardation, edemas, retarded swim bladder inflation, and hyperactive swimming behavior [22]. Microphthalmia was also observed as a striking malformation, probably due to impaired function of the condensin I complex involved in chromosome segregation during mitosis.

One of the lesser studied issues in environmental risk assessment is whether POPs can cause deleterious effects on bone and cartilage development. Indeed, their diverse effects on general metabolism (e.g. as endocrine disruptors) carry the risk of affecting skeletal development (scoliosis, craniofacial) as well as pathologies such as osteoporosis or osteoarthritis [23]. Some POPs, such as polychlorinated biphenyls (PCBs), pesticides and dioxins, have been found in connective tissue such as cartilage and bone in several species where its uptake can be traced [23–25]. Some studies suggest a direct deleterious effect on the function of chondrocytes after exposure to POPs [26]. It has been hypothesized that these POPs could cause damage to such tissues by disrupting the balance between cartilage formation and degradation, which could lead to its breakdown and consequently to the development of osteoarthritis [27]. Chondrocyte and osteoblast malfunctions may result from several factors such as gene mutations [28, 29], environmental stress causing shifts of the glycolytic pathway [30], age-related effects, and even sex hormone deficiencies [31].

For our study we have chosen the zebrafish larvae as our testing model. This model organism has received extra attention as attempts have been made to reduce the use of animals for experiments. The zebrafish's popularity has increased in the past years due to its many advantages, such as a high degree of similarity in the genome relative to humans, the lower operational costs compared to other models, their capacity to produce often and numerous

offspring [32]. This species has been used to model diseases, genetic conditions, effects of pollutants, and many more across different disciplines. Furthermore, it is a promising model to test environmental chemicals and craniofacial skeletal development [33–35].

Here we looked at the morphological defects, specifically those observed in bone and cartilage of zebrafish larvae after exposure to an environmentally relevant POP mixture. Furthermore, we analyzed the transcriptome to obtain clues about the mechanisms involved, and through cluster analysis, in an attempt to elucidate the potential binding mechanisms, we also compared the compounds using their structural properties with model (ant)agonists.

## Results

### POP exposure led to craniofacial alterations and severe disruption of the chondrocranium morphology

Previous experiments have tested the total mixture of 29 POPs on zebrafish larvae [17], where a lethal concentration of LC50 = 386-fold the mean human blood concentration (POP386×) was found [22], while exposure to POP75× or POP125× resulted in more than 95% survival at 4 dpf. On the other hand, POP125× did affect behavior, heart rate, and eye development [22] and can be considered as a realistic scenario in highly exposed populations (*e.g.* sea food, environmental/natural disaster). Thus, the POP125× mix (S1 Table) was chosen to expose WT zebrafish larvae during the first 4 days for studying the effects on skeletal development.

At 5dpf, the larvae were stained with alcian blue (AB) to reveal the cranial cartilage elements. Morphometric measures were performed on these treated larvae, revealing a significant decrease in the angle between ceratohyals (ACT) and in the distance between the left and right Meckel's cartilage/palatoquadrates (MPQ) (Fig 1A and 1B). In contrast, longitudinal measures such as the length of the ceratohyals (LC) or the distance between the frontal end of the ceratohyals and the line connecting the posterior ends of the hyosymplectics (DCH) were unchanged (Fig 1C and 1D), while the combined distance covering the entire Meckel's cartilage, palatoquadrate, and the hyosymplectics (MPH) was significantly decreased (Fig 1E). Thus, globally the length of the head cartilage seems unaffected by POP125× treatment, while the medial region of the skeleton appears to be narrower. This is also illustrated by the inward curving of the palatoquadrate, leading to an extremely narrow angle in its connection to the hyosymplectics (arrow in Fig 1F). This deformity was observed in around 50–60% of the treated animals.

To assess which one of the components of the POP mix was responsible for the observed defects, different sub-mixes (PFAA, Br, Cl) at equivalent 125× concentrations were tested, as well as their dual combinations (PFAA+Br, PFAA+Cl, Br+Cl) in independent experiments (Fig 1). A significant decrease in the angle between ceratohyals (ACT) was observed only for PFAA+Br and PFAA+Cl, similar to the distance between Meckel's cartilage and palatoquadrate (MPQ), with PFAA+Cl being the most effective ($p < 0.001$). The length from the frontal part of the ceratohyals to the back of the hyosymplectic (DCH) was unaffected by any of the treatments. The length of the ceratohyal (LC) was altered in groups such as PFAA and PFAA+Cl, but not PFAA+Br, while the distance covering hyosymplectic, palatoquadrate and Meckel's cartilage (MPH) was decreased by all treatments containing PFAA within their formulation. We also evaluated the differences between PFAA alone and the other treatments. Only three observed parameters had significant differences: the width of the angle (ACT) where POP125×, PFAA+Br and PFAA+Cl had smaller values than PFAA, and in both MPH and MPQ where the measured length was larger in Br than PFAA.

To further assess the occurrence of craniofacial deformities, each fish was observed, then assessed, and catalogued. Three categories were established based on the severity of the skull

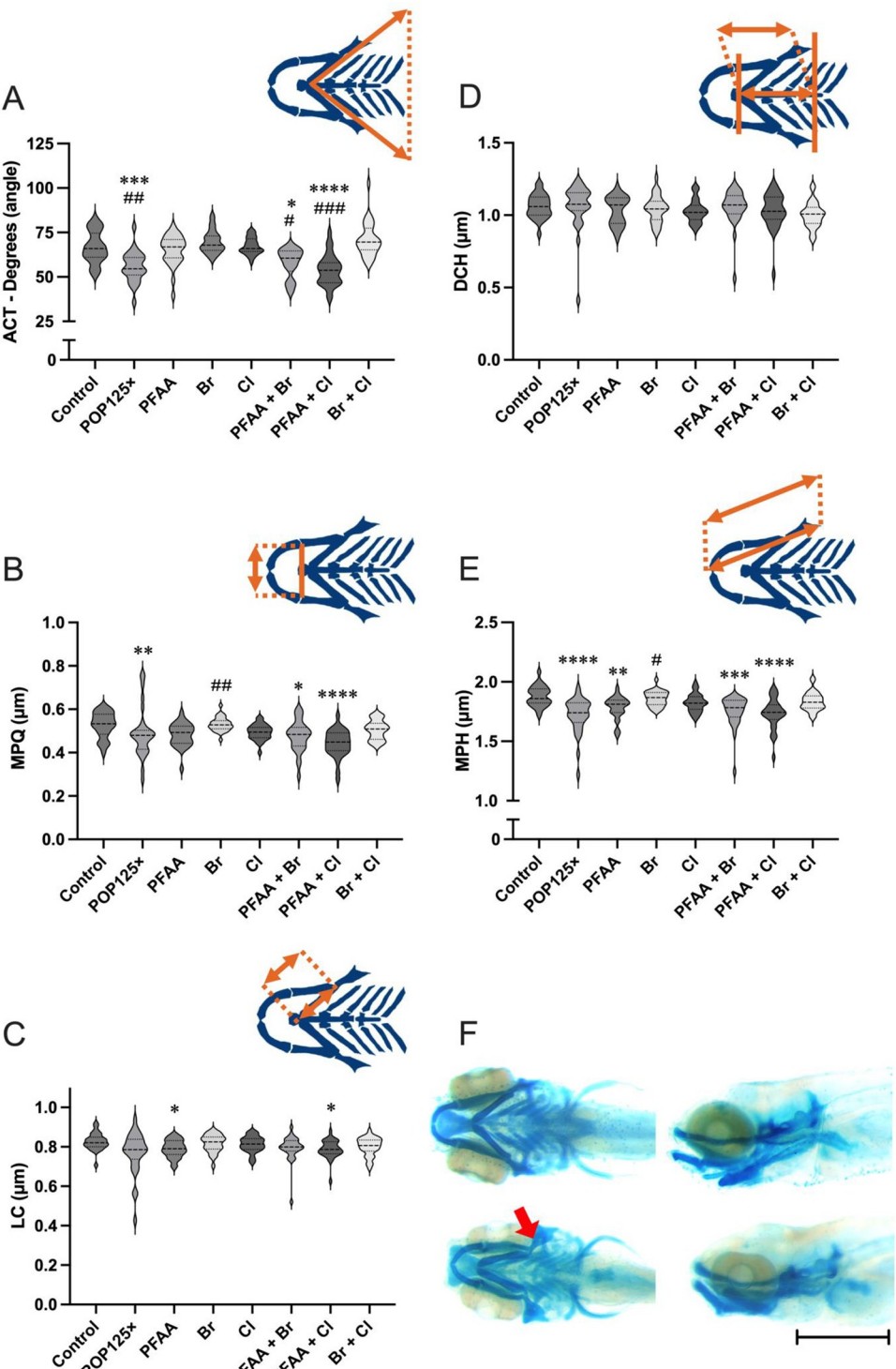

**Fig 1. Morphometric analysis of the chondrocranium in 5dpf zebrafish larvae exposed to POP mixtures.** A) Aperture of the angle between ceratohyals (ACT), B) Distance between the left and right Meckel's cartilages/ palatoquadrates (MPQ), C) Length of the Ceratohyal (LC), D) Distance between the frontal end of the ceratohyals and the line connecting the posterior ends of the hyosymplectics (DCH); E) Combined distance covering the entire Meckel's cartilage, palatoquadrate, and the hyosymplectics (MPH); F) Alcian blue staining of controls (upper pictures) and larvae exposed to POP125× (Bottom pictures), bar represents 1 mm. Kruskal-Wallis with Dunn's multiple comparison test. $p < 0.05$ (*), $\leq 0.01$ (**), $\leq 0.001$ (***), $< 0.0001$ (****). Asterisks (*) when differences were found compared to Control, pound sign (#) when differences were found compared to PFAA.

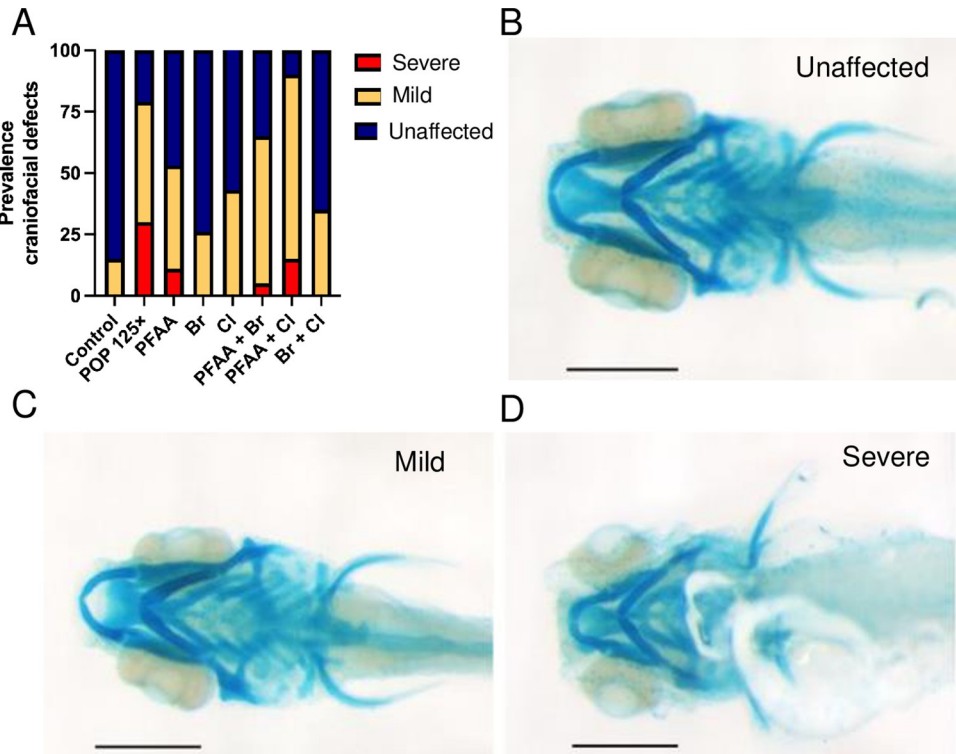

**Fig 2. Prevalence of craniofacial defects and examples of such defects.** (A) Percentage of fish presenting craniofacial defects, (B-D) Examples of phenotypes used to catalogue fish in each category: (B) normal, (C) mild, (D) severe phenotype. The scale bar represents 200 μm.

defects. For this analysis, the shape of the different structures composing the head was qualitatively assessed, then assigned to one of the three categories, namely "Unaffected", "Mild", and "Severe" (Fig 2 and S2 Table). In the "Unaffected" category, only Br and Br+Cl revealed no significant differences, while all other treatments resulted in significant decreases compared to control. This translated in a slight, but significant increase in "Mild" phenotypes for the Cl treatment, in contrast to the very significant increases by those formulations where PFAA was present, that is POP125×, PFAA, PFAA+Br, PFAA+Cl. "Severe" phenotypes were only found in the PFAA-containing groups, but only POP125× reached significance due to a high variability. Finally, comparing all exposure groups containing PFAA vs PFAA alone, the POP125× and the PFAA+Cl treatments showed a significantly lower incidence of the "normal" category, while PFAA+Br caused a significant increase in the 'mild' category.

## Mineralized bone formation is compromised following 4-day continuous exposure

To assess the effect of POPs on bone mineralization, we performed alizarin red staining on 10 dpf fixed larvae after treatments, as at this stage most cranial bone elements are already mineralizing [36, 37]. After the different treatments from day 0 to day 4, the larvae were grown until 10dpf before staining them using a traditional staining protocol (euthanize then stain). Unfortunately, no survivors were obtained at this stage from the larvae treated with POP125×, PFAA and PFAA+Cl. The area of the opercle was measured, as it is the most prominent structure at this stage (Fig 3). In surviving individuals treated with PFAA+Br, the area of the opercle was significantly smaller compared to controls and those subjected to other treatments. The

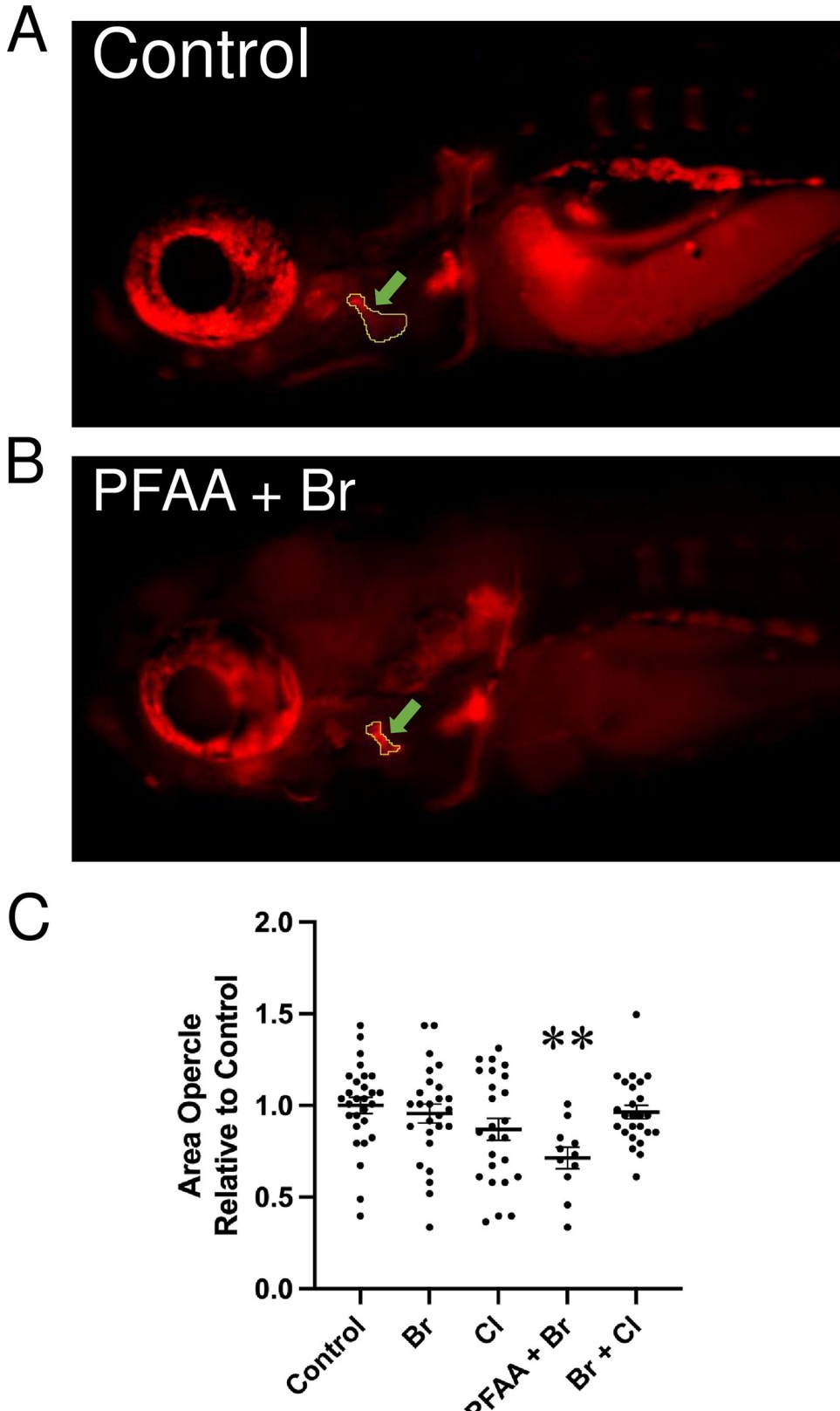

**Fig 3. Area of the opercle in fish stained with alizarin red at 10 dpf.** (A, B) Live alizarin red stained fluorescent image of a control larva (A) and a larva treated with PFAA+Br (B) (lateral view, anterior to the left). Arrows point at

the area of the opercle outlined in the images. The scale bar represents 500 μm. (C) Opercle area measured in control larvae and larvae upon treatment with the indicated mixture ($n \geq 11$). Ordinary one-way ANOVA and Dunnett's multiple comparison test were performed. Significant difference relative to control is indicated: p-value<0.05 (*), $p$ value $\leq 0.005$ (**).

mixtures Br+Cl and Br alone did not have any remarkable effect. On the contrary, Cl treated fish had slightly smaller area sizes, but not enough to be considered as significantly different from controls.

Following the assessment of bone mineralization, in a second experiment, the transgenic line *Tg(col10a1a:col10a1a-GFP)* was used, which contains the coding sequence for the fluorescent protein GFP inserted into the coding region of the endogenous *col10a1a* gene, thereby causing a fusion protein *col10a1a-GFP* mRNA to be expressed under the control of the endogenous *col10a1a* promoter [38]. The expressed fusion protein is secreted from the cells and then binds to mineralized or un-mineralized (total) bone matrix in the living larvae. To detect specifically mineralized bone structures in the same individuals, the larvae were stained live with alizarin red for red fluorescence detection. The green fluorescence revealed a significant decrease in the total bone matrix only upon treatment with Br and PFAA+Br (Fig 4A). Surprisingly, no significant difference was observed in the live staining of mineralized bone between all the treatments, possibly due to the longer manipulation time compared to the direct fixation used above (Fig 4B). The obtained ratio between total and mineralized bone matrix, only PFAA+Br revealed a significant decrease in the opercle area (Fig 4C). Further, this analysis indicates that mainly the PFAA+Br mixture causes a significant decrease in mineralized bone formation, with a major effect on deposition of the unmineralized bone matrix.

## Modulation of expression of genes related to skeletal development

Differentially expressed genes (DEGs) upon POP75× and POP125× treatment were previously analyzed by whole larvae, whole transcriptome RNA-Seq [22](the data are accessible through GEO Series accession number GSE208019 (https://www.ncbi.nlm.nih.gov/geo/query/acc.cgi?acc=GSE208019)). In light of the defects in skeletal development reported here, the list of DEGs was reanalyzed according to their known expression pattern, as available from zfin-org. Interestingly, this analysis revealed a significant enrichment in down-regulated genes that are expressed in the pharyngeal arch 3–7 skeleton and the splanchnocranium (visceral head skeleton), while only GSEA identified the pectoral fin bud (S3 Table). In addition, the functional enrichment analysis of DEGs previously carried out was reconsidered (S3 Table). In molecular functions, many membrane receptors appeared to be affected in their expression and signaling, while one class of nuclear, ligand-regulated receptors was striking. Perturbations of the vitamin D and retinoic acid pathways were shown to affect skeletal development and may lead to malformations of the cranium [39–47]. Indeed, the genes coding for vitamin D3 receptor (*vdra*), retinoic acid receptor (*rargb*), and peroxisome proliferator-activated receptor (*pparda*) were all significantly upregulated ((log2(fold-change) of 1.90, 1.95, and 1.82, respectively) as well as that for their common heterodimerization partner Rxr (*rxrab*: log2(fold-change) = 1.79). Using Cytoscape, a network of these zebrafish genes (S3 Table) was constructed, based mainly on their shared protein domains (Fig 5), while a similar analysis based on their human homologs revealed a dense network of genes linked by known physical, genetic and signaling pathway interactions of the encoded proteins. It was observed that the *rxrab*, *vdra*, *vdrb*, *rargb* and *pparda* genes are upregulated in a dose-dependent manner upon exposure to POP75× and POP125×. In that context, it is also interesting to mention that the gene *cyp26c1*, coding for an enzyme involved in retinoic acid degradation and regulating bone mineralization [39–47] is

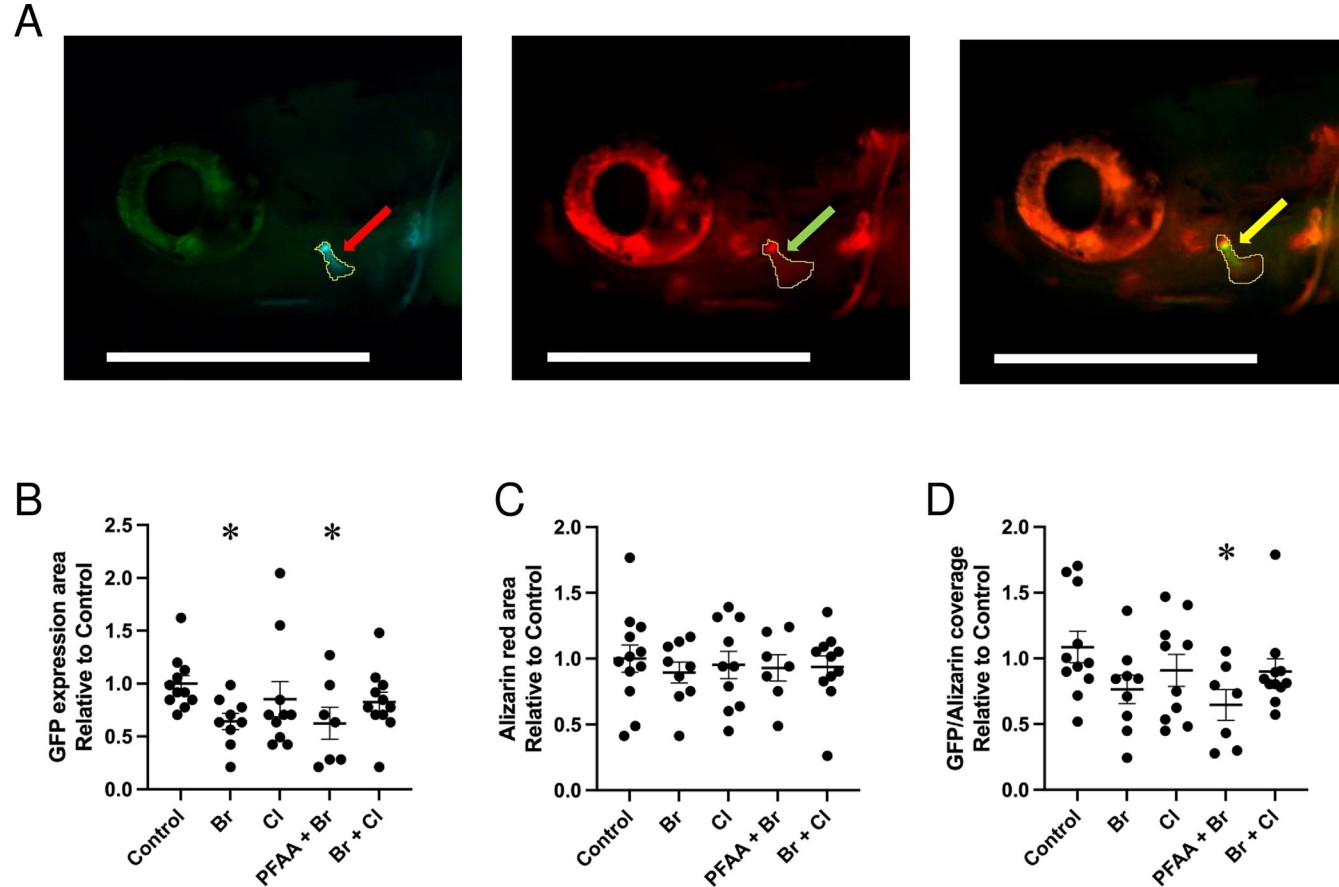

**Fig 4. GFP staining in *Tg(col10a1a:col10a1a-GFP)* and alizarin red staining of the opercle in 10 dpf zebrafish.** (A) Example of an individual control larva illustrating the green fluorescence of the *col10a1a-GFP* fusion protein (left), the red fluorescence of the live alizarin red staining (middle) and an overlay of both (right). The outline of the fluorescent areas is shown (red, green, and yellow arrows). Lateral view, anterior to the left, the scale bar represents 500 μm; (B) Plot of the opercle area revealed by the green fluorescent *col10a1a-GFP* fusion protein; (C) Plot of the opercle area stained with alizarin red D) Plot of the ratio between the green fluorescent and red fluorescent opercle area measured in control larvae and larvae upon treatment with the indicated mixture (n ≥ 7). Ordinary one-way ANOVA and Dunnett's multiple comparison test were performed. Significant difference relative to control is indicated: *p* value < 0.05 (*).

upregulated upon POP treatment (log2(fold-change) = 0.8; p-value = 0.006) (S4 Table). The *cyp2r1* gene, involved in vitamin D biosynthesis, is downregulated, while the gene for vitamin D degrading enzymes *cyp24a1* is significantly upregulated [48].

Another interesting molecular function and Reactome pathway was that for collagen biosynthesis and modifying enzymes, which appeared to be preferentially downregulated, pointing to defects in the extracellular matrix affecting cartilage and bone formation. These genes aggregate into two connected networks linked by shared protein domains and coexpression in zebrafish, while their human homologs form a dense network of downregulated genes whose encoded proteins are linked through physical, co-expression, co-localization, and genetic interactions (Fig 6). Interestingly, all these genes are downregulated in a dose-dependent manner upon exposure to POP75× and POP125× treatment. The different collagen genes form a specific cluster of related genes, while another cluster is formed by genes coding for enzymes involved in collagen maturation. The *plod3* gene codes for a procollagen-lysine, 2-oxoglutarate 5-dioxygenase 3 enzyme whose human homolog causes craniofacial deformities and bone fragility when mutated [49]. Similarly, the *p3h1*, whose human ortholog is involved in *osteogenesis imperfecta* [50] and *p3h3* genes code for prolyl-3-hydroxylases, while the *p4ha1a* and the

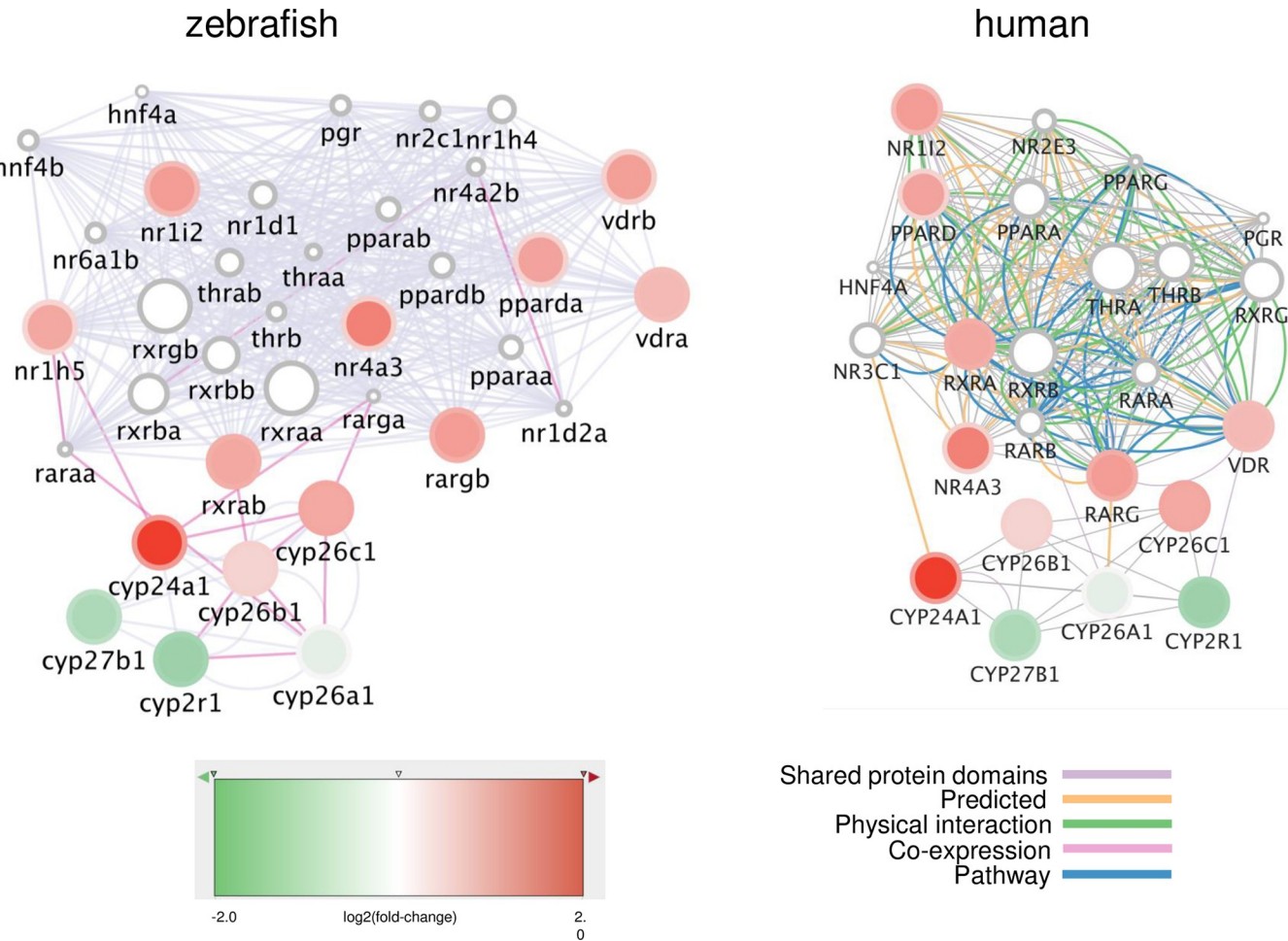

**Fig 5. Networks of genes coding for nuclear receptors differentially expressed upon exposure to the total POP mixture.** Networks were generated on Cytomine using GENEMANIA annotations. The nodes represent genes, the fill colors indicate the log2(fold-change) upon POP125× exposure, while the rim circle color gives the log2(fold-induction) due to POP75× exposure. All genes are upregulated in a dose-dependent way. The edges (connecting lines) represent links between nodes, based on "Shared protein domains" in the zebrafish network (left), or on specific interactions and pathways as indicated for the human network (right).

*p4ha2* genes code for prolyl-4-hydroxylases whose mutation in mouse causes impaired ECM, chondrodysplasia, and kyphosis [51].

## Chemical similarities between POPs and retinoids or vitamin D

In an attempt to connect chemical structures of pollutants to specific pathways they may affect, the structural similarity of the molecules within the POP mix with some 'model' compounds known to act on specific pathways was evaluated. We decided to focus on the signaling that we found to be affected by the POPs and who are known to affect skeletal development, the vitamin D and the retinoic acid pathways. Agonists for the Vdr (calcitriol) or Rar (retinoic acid) were used as reference compounds and their chemical similarities were computed to all compounds in the POP mix. In addition, 4 triazole fungicides with a potential to be endocrine disruptors and posing serious concerns were chosen to complete the analysis: flusilazole, triadimenol, diniconazole and hexaconazole [52–59]. Hierarchical agglomerative nesting clustering was performed using the chemical fingerprints (toxprints) of all these compounds (Fig 7).

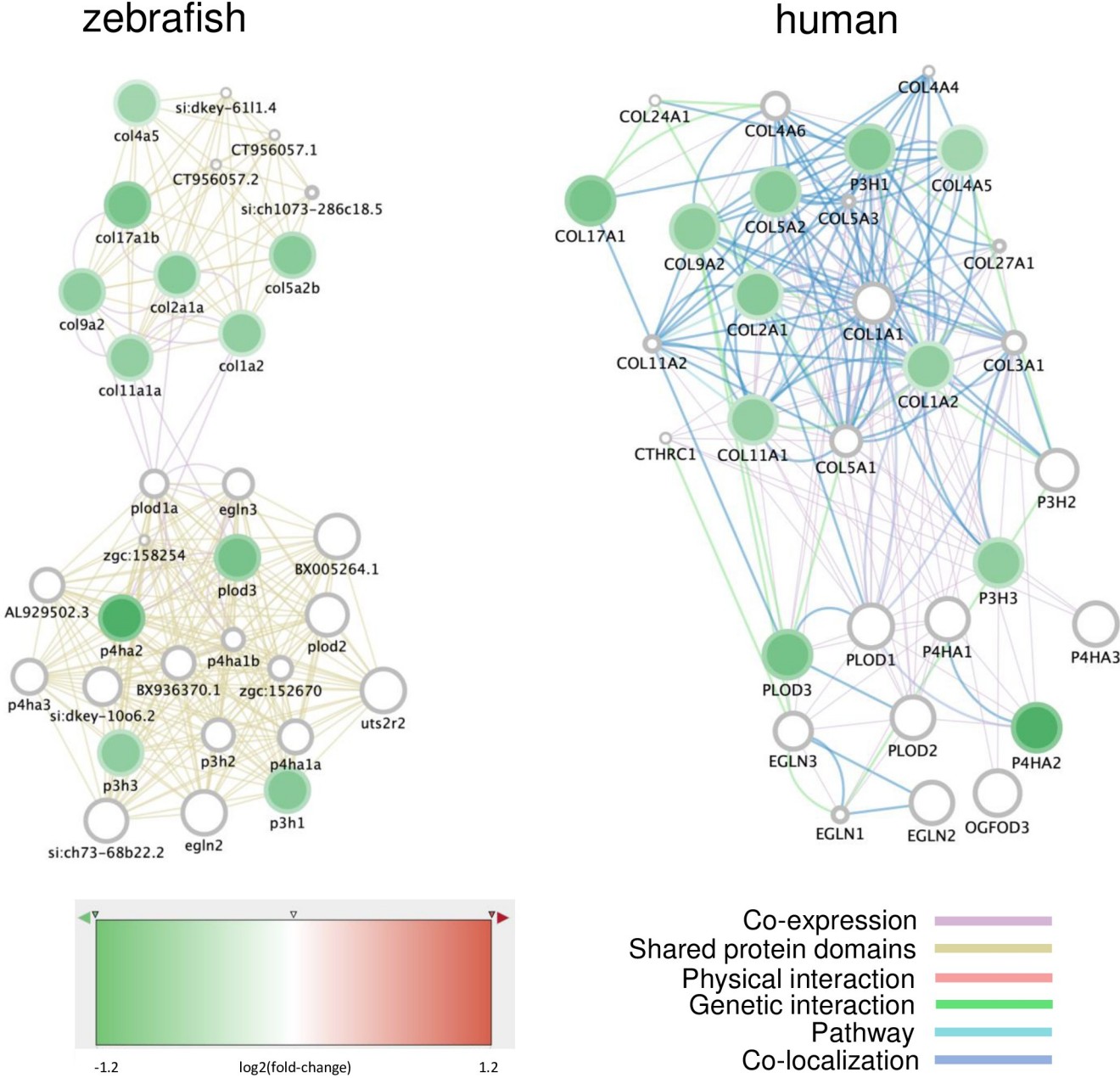

**Fig 6. Networks of genes involved in collagen synthesis and differentially expressed upon exposure to the total POP mixture.** Networks were generated on Cytomine using GENEMANIA annotations. The nodes represent genes, the fill colors indicate the log2(fold-change) upon POP125× exposure, while the rim circle color gives the log2(fold-induction) due to POP75× exposure. All genes are down-regulated in a dose-dependent way. The edges (connecting lines) represent links between nodes, based on "Shared protein domains" and "Co-expression" in the zebrafish network (left), or on specific interactions and pathways as indicated for the human network (right).

As expected, closely related compounds such as polychlorinated biphenyls (PCBs), brominated flame retardants (PBDEs), and perfluoroalkyl acids (PFAAs) clustered together, indicating that the applied method is successful in grouping chemicals (Fig 7). Interestingly, the compounds arranged in two main clusters, with an agglomerative coefficient of 0.9568738. The first cluster (orange box) is headed by calcitriol, with the nearest relatives dieldrin and

| 1 | Calcitriol |
|---|---|
| 5 | Diniconazole |
| 35 | Dieldrin |
| 3 | Flusilazole |
| 19 | HBCD |
| 28 | HCB |
| 6 | Hexaconazole |
| 27 | p,p'-DDE |
| 30 | Oxy-chlordane |
| 13 | PBDE-47 |
| 14 | PBDE-99 |
| 18 | PBDE-209 |
| 16 | PBDE-153 |
| 17 | PBDE-154 |
| 15 | PBDE-100 |
| 20 | PCB-28 |
| 21 | PCB-52 |
| 22 | PCB-101 |
| 23 | PCB-118 |
| 24 | PCB-138 |
| 25 | PCB-153 |
| 26 | PCB-180 |
| 2 | Retinoic Acid |
| 31 | Trans-nonachlor |
| 4 | Triadimenol |
| 29 | α-chlordane |
| 32 | α-HCH |
| 33 | β-HCH |
| 34 | γ-HCH |
| 7 | PFHxS |
| 11 | PFDA |
| 8 | PFOS |
| 9 | PFOA |
| 10 | PFNA |
| 12 | PFUnDA |

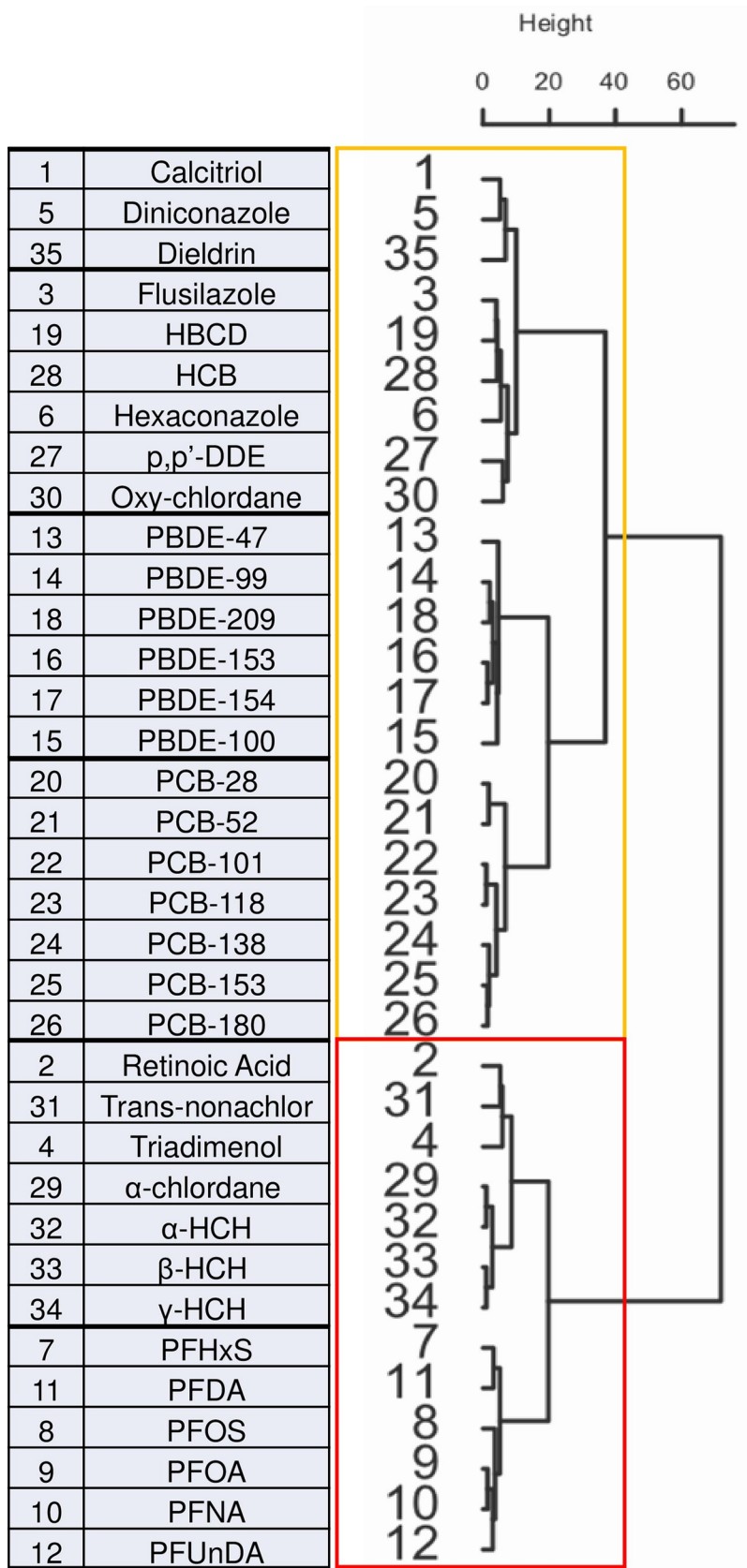

**Fig 7. Molecular fingerprints organized in hierarchical agglomerative nesting clusters.** Dendrogram revealing the chemical fingerprint similarity relations between reference compounds and the components of the POP mixture (29 compounds).

diniconazole, followed by HBCD, HCB, $p,p'$-DDE, flusilazole hexaconazole, and oxy-chlordane, while the remaining PCBs and PBDEs sub-clusters were connected more distantly.

The second clearly distinct cluster (red box) grouped chemicals more similar to all-trans-retinoic acid (ATRA), with trans-nonachlor and triadimenol as the closest relatives followed by the organochlorinated pesticides α-chlordane and HCH, and finally the PFAA cluster.

Taken together, this clustering experiment shows that it is possible to classify compounds based on their chemical fingerprints, while also giving an indication on their possible target pathway through which they may exert their effects. In the pollutant list considered here, some compounds are more likely to act through binding to the Vdr, while others would preferentially act on the retinoic acid pathway.

## Discussion

A lot of research effort during the last decades has backed up the affirmation that environmental pollutants, and specifically POPs are a real threat to both humans and the environment. Most of the studies on environmental toxicity use standardized protocols to evaluate lethal, morphological, or reproductive effects, while human health studies focus on carcinogenicity, genotoxicity, or adverse effects on specific organs such as liver, brain, or reproductive organs as also investigated by the pharmacological industry during the drug discovery process [60].

This work focused on the potential of POPs to induce defects in the skeleton of developing vertebrates, using the zebrafish larva as a convenient and sensitive model system. Moreover, the fact that POPs are nearly exclusively found as mixtures [13–16] was considered, thus limiting the interpretation of the biological relevance for health and environment of studies, though insightful, focusing on single compounds. Therefore, a mixture of POPs was tested based on the concentrations found in the blood of a Scandinavian population [17] and sub-mixtures thereof. Skeletal deformities in the chondrocranium were observed as well as decreased bone mineralization in 5dpf larvae exposed to the POP. The observed malformations complement the effects on growth, heartbeat, behavior, and eye development that were previously reported [22].

This is the first time such an extensive evaluation of skeletal effects has been conducted using a realistic mixture of POPs that we are aware of. On the one hand, the effects on morphology of the head cartilage elements, potentially leading to craniofacial deformities in the adult, were investigated, and bone mineralization was assessed using the growing opercle as one of the most prominent bones to be observable. To some extent, altering the normal function and viability of chondrocytes could lead to alteration of the osteoblast activity and the subsequent formation of bone [26], however the opercle is an intramembranous bone which does not depend on a preformed cartilage matrix to develop. Nevertheless, many regulators and signaling pathways are common to both processes, thus some of the mechanisms involved in deficiencies may be shared.

Some of the compounds within the mixture have been tested previously as single compounds. PCBs such as PCB 28, PCB153, and the flame retardant metabolite (6-OH-BDE-47) were found to cause incomplete fusion of the ethmoid plate and reduced size of the jaw and branchial cartilages in zebrafish [61, 62]. *In vitro* experiments using the murine chondrogenic ATDC-5 cell line and human T/C-28a2 immortalized chondrocytes that were exposed to non-dioxin-like PCBs such as PCB 101, PCB 153 and PCB 180 showed that these compounds induce chondrocyte apoptosis after exposure [27]. In humans, PCB exposure might be linked

to rheumatoid arthritis and osteroarthritis [63], while a recent overview [26] reported possible associations of PCB and PFAA contaminations with increased incidence of osteoarthritis in human cohorts.

Osteotoxicity has been witnessed across species following exposure to one or several of the chemicals within the mixtures used here. For instance, bone density in polar bears has been negatively correlated to the presence of POPs such as *p,p′*-DDE, HCH and PBDE-153 [64]. In humans, exposure to organochlorine compounds has also produced changes in bone mineral density [65, 66]. Some of these effects can even be witnessed across generations and through maternal exposure. The offspring of female goats orally exposed to PCB 153 had alterations in the bone composition following exposure [67]. Perfluoroalkyl substances such as Perfluorooctanoic acid (PFOA), Perfluorooctane sulfonate (PFOS), Perfluorononanoic acid (PFNA) and Perfluorohexane sulfonate (PFHxS) have been found in human bones and, since they tend to accumulate, they have the potential to affect bone turnover, hence altering bone geometry and mineral density. Uptake of PFAS by osteoclasts was seen in *in vitro* experiments, and PFNA was present within human bones [68–70]. The presence of PFOA [71] or other PFAAs [68, 72–74] in human blood serum has been associated with lower bone density, changes in cell differentiation, and bone weakness through several stages in life. Some PFAAs, such as PFOA, have been shown to bind peroxisome proliferator-activated receptors (PPAR) receptors, whose deregulation can lead to metabolic disorders and contribute to bone defects [3, 68, 73, 74].

Mild to severe effects on cartilage development were observed in 5dpf larvae caused by the different treatments, with the total mix POP125× being the most deleterious. Sub-mixes Br, Cl, and Br+Cl, were in general not causing very significant defects, only PFAAs alone caused a significant increase in the "mild" phenotype. The dual mixtures PFAA+Br and PFAA+Cl caused significantly higher percentages of malformations, close to or even exceeding those observed with the total mix. Thus, it appears that the significant malformations caused by the total POP125× mix results from largely cumulative actions of each of the sub-mixes, with a stronger effect caused by PFAAs.

A similar picture emerges concerning bone mineralization of the opercle in 10dpf larvae, although here the POP125× mix, PFAA, and PFAA+Cl could not be assessed due to their lethality at this stage. However, a decrease in mineralization upon PFAA+Br treatment was observed. Moreover, a decrease in the total bone matrix (as assessed by staining by the transgenic fusion protein Col10a1a:GFP) was observed, caused by the Br and the PFAA+Br mixtures, as well as a decrease in the ratio between total bone matrix/mineralized bone matrix caused by PFAA+Br (Fig 4D). The discrepancy between the two alizarin red staining experiments, one performed directly on fixed WT larvae (Fig 3), the other by live staining of living larvae, is probably due to the different timing and protocol of the observation and/or slight differences in the development of the WT and the transgenic line. Taken together, these observations indicate that bone formation is primarily decreased by interfering with the capacity of osteoblasts to deposit the bone extracellular matrix. This conclusion is supported by the transcriptomic analysis of 5dpf larvae treated with POP75× or POP125× mixtures which clearly identify a dose-dependent decrease of collagens and collagen maturation enzyme genes upon exposure (Fig 5).

Many hormones, such as parathyroid hormone (PTH) [39], 17-α-ethinylestradiol (EE$_2$) and 17-β-estradiol (E$_2$) [75, 76], as well as environmental pollutants such as 2,3,7,8-Tetrachlorodibenzo-p-dioxin (TCDD, dioxin) [77] have been shown to induce malformations such as bent palatoquadrate cartilages, shorter ceratohyal cartilages, changes in the angle of the Meckel's cartilage and even some missing structures, while benzo (*a*) pyrene (BaP) was shown to affect the expression of several skeletal genes (*sox9a*, *spp1*, *col1a1*) [78]. However, the POP125× mixture used here does not contain, on purpose, any dioxin-like compound (PCBs),

nor BaP [17]. Previous studies showed by RNA-Seq analysis that this mixture does not present estrogen agonistic or antagonistic properties as none of the classical target genes for the estrogenic pathway were found to be affected [22]. Similarly, PTH-like compounds would affect calcium homeostasis for which no relevant gene was detected. In contrast, down-regulation of target genes for the androgen receptor was observed, while anti-androgenic properties of the POP-mix were previously shown in a cell-based assay [18]. Androgens are known to be required for osteoblast differentiation and bone growth [79], it is thus possible that interference with androgen signaling is causing some of the observed effects. However, this transcriptomic analysis revealed in addition a dose-dependent increase in the expression of the *vdra*, *vdrb*, the *rarga*, *rargb*, the *rxrab*, and the *ppard* genes. Ppar receptors are mainly involved in lipid metabolism, however effects of Pparb/d on skeletal health have been shown [80], and PFOA was revealed as an agonist of the PPARa receptor, hence possibly directly affecting bone homeostasis [71]. Transcriptomic analysis further revealed that regulation of genes coding for enzymes involved in vitamin D and retinoic acid metabolism support an increase of these signaling pathways, which are well known for their involvement in skeletal formation. Vitamin D is well known for its role in preserving bone integrity in humans [81], a *vdra* deficient zebrafish presented delayed vertebral ossification, while treatment with exogenous vitamin D was shown to cause cranial skeleton deformities in developing zebrafish larvae [39]. Retinoic acid similarly has well described effects on bone formation and homeostasis in humans [82], while its effects in zebrafish have been well studied in individuals deficient in the RA degrading enzyme Cyp26b1, causing increased RA levels and severe developmental and craniofacial deformities [40, 42, 44]. Interestingly, comparison of the chemical fingerprints with the *bona fide* ligands revealed a clear separation into compounds more related to vitamin D and those more similar to retinoic acid. Taken together, these considerations support a hypothesis that disruption of one, or several of these pathways may, at least in part, be involved in the skeletal defects caused by the POP mix.

This study also illustrates the extent to which using specific transgenic lines, coupled to live staining techniques that are available for performing studies in zebrafish larvae can help to identify developmental toxicity rapidly and easily, as well as mechanisms of action and potential molecular targets of individual compounds and mixtures. A decrease in mineralized bone formation, as observed here, may be due to inhibition of osteoblast proliferation, differentiation, inhibition of ECM deposition, or the final step of mineralization (Fig 8) [36, 83, 84]. Here, the transgenic line *Tg(col10a1a:col10a1a-GFP)* was used to reveal total (i. e. unmineralized and mineralized) bone ECM, combined with live staining of the mineralized ECM [38]. This experiment revealed that the POP125× more strongly affected deposition of the unmineralized ECM than the subsequent mineralization. This is consistent with the observed decrease in expression of the collagen and collagen maturation enzyme genes. Furthermore, manually checking the RNA-Seq results from treated larvae [22] did not reveal any changes in expression of genes involved in osteoblast differentiation, such as *sox9a*, *runx2b*, *sp7*, or *spp1* [85]. Similarly, genes involved in inflammation or oxidative stress (*sod*, *gsh*) were not significantly affected. In conclusion, our results strongly suggest that mainly the deposition of collagenous bone ECM is affected by the POP125× mixture tested here in 5 dpf zebrafish larvae (Fig 8).

Molecular fingerprints of a compound are based on substructure keys that are used to search for structure similarity, representing different aspects of a molecule [86]. This similarity is then used to propose toxicity alerts based on these chemical features, common scaffolds, and varied ring, bond, and atom types. This kind of approaches have been used for drug discovery and virtual screening. Here, structural similarity was applied to identify the chemicals most likely to bind to and exert effects on receptors such as retinoic acid, vitamin D, and *pparda*, as suggested from our transcriptomic analysis. The chemical fingerprints approach used here,

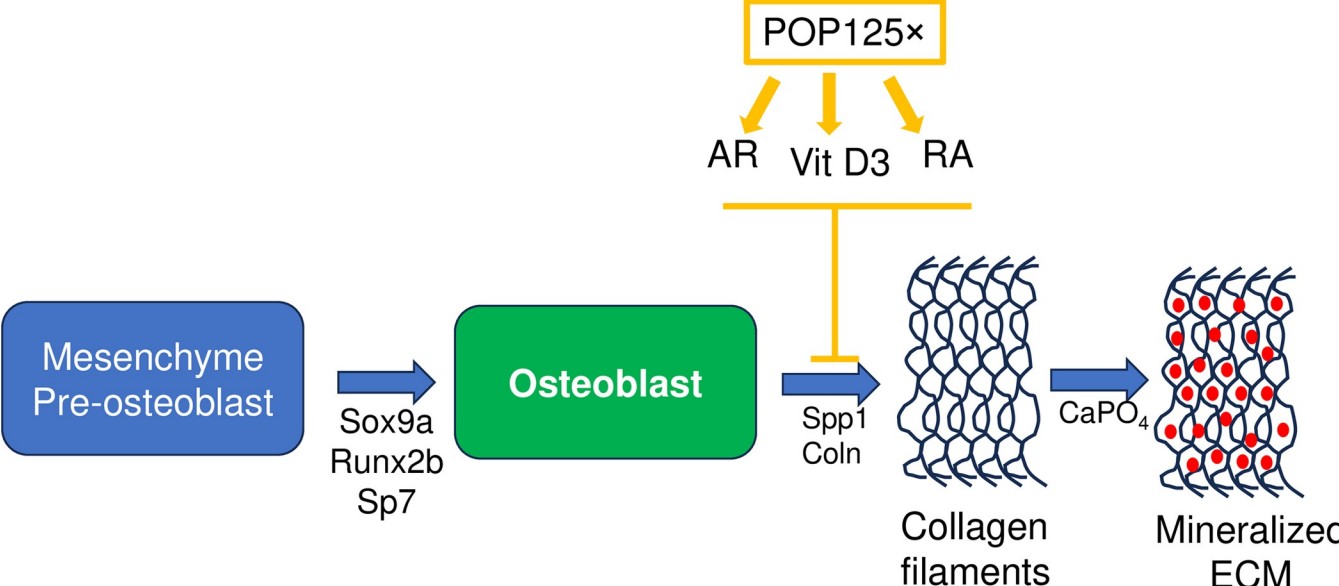

**Fig 8. Schematic diagram of the effects of POP125× on skeletal development.** Represented is the bone-forming osteoblast cell, originating from pre-osteoblastic, mesenchymal cells through expression of regulatory genes such as *runx2b* or *sp7*. They secrete the collagenous (Coln) bone ECM, which itself is subsequently mineralized through deposition of $CaPO_4$ cristals. POP125× inhibits mineralized bone formation primarily by interfering with collagen secretion, acting on the vitamin D (vit D3), the retinoic acid (RA), and/or the androgen receptor (AR) pathways.

and other similar ones also based on structural similarity of the compounds [87], may be useful to suggest potential mechanisms of action of a compound, and it could aid in focusing on specific components of a mixture to reduce the experimental testing load. Thus, using chemoinformatic data has major potential implications in reducing specifically animal testing, while dealing with the tremendous task of evaluating risks caused by mixtures. Further work is needed to confirm and improve such a strategy, also including other kinds of properties (*e.g.*, toxicity, environmental levels, bioavailability), however it does have potential implications for regulatory agencies, both dealing both with environmental and human health regulations, that are moving towards non-testing studies, as laid out for example in the New Approach Methodologies [88] for human risk assessment.

Taken together, we propose that a realistic mixture of POPs, as derived from the blood of a Scandinavian population, causes skeletal malformations and decreased bone formation, most likely through affecting the vitamin D and retinoic acid signaling pathways. Our study helps to understand the potential effects caused by exposure to a realistic mixture of POPs, an exposure scenario that could be reached in particular populations (or subgroups within a population) of humans or wildlife, especially those living by the ocean and relying heavily on fish as a food source [89–92].

## Materials and methods

### Zebrafish husbandry and ethical considerations

Adult wild-type zebrafish (WT) of the AB strain and *Tg(col10a1a:col10a1a-GFP)* [38] were obtained from breeding facilities at the GIGA-Institute, Liege, Belgium. Fish maintenance, breeding conditions, and egg production were described [93, 94] and are in accordance with internationally accepted standards. Animal care and all experimentation were conducted in

compliance with Belgian and European laws (Authorization: LA1610002 Ethical commission protocol ULg19-2134 and ULg19-2135).

## Chemicals, persistent organic pollutant mixtures and exposure tests

Dimethyl sulfoxide (DMSO, >99.9%, CAS number 67-68-5) was purchased from Sigma-Aldrich (Merck KGaA, Darmstadt, Germany). The stock solutions for total POP mixture and six sub-mixtures were designed and prepared by the Norwegian University of Life Sciences, Oslo, Norway [17] as indicated in S1 Table. Briefly, the stock solution of the total POP mixture was designed to represent a mixture of 29 compounds at 1,000,000-fold the mean concentrations found in the blood of a Scandinavian population, while the sub-mixtures consisted of the same concentrations of either one single class of these compounds (PFAA, Br, Cl) or of two combined (PFAA+Br, PFAA+Cl, Br+Cl) classes. The stock solutions were diluted in DMSO to obtain a final concentration of 125-fold the mean concentrations found in the blood of a Scandinavian population (125×) [22]. Exposure tests were performed in 6 well-plates, with 25 fertilized eggs per well in 4 mL of E3 medium supplemented or not with the test compounds. DMSO concentrations were 0.1%. For each experiment, 150 fertilized eggs were selected, 50 as controls and 100 for the specific treatment, to ensure a sufficient number of treated individuals for the tests. Each treatment was repeated at least three times in independent experiments. To keep stable chemical concentrations, we used a static-renewal approach where at least 90% of the media was refreshed every 24 h. Exposure started between 0 to 6 h post fertilization (hpf), the larvae were treated for 96 h before being transferred to fresh E3 medium without compounds for further growth for one or 6 days.

## Craniofacial morphometrics

Cartilage integrity was assessed in 10 wildtype zebrafish larvae per treatment that were staged, euthanized with an overdose of tricaine (400 mg/L) (MS-222, Ethyl 3-aminobenzoate methane sulfonate; Merck, Overijse), fixed in PFA 4% overnight (ON) at 4°C and stained with alcian blue (Sigma-Aldrich/Merck, Overijse, Belgium) solution (EtOH 80%/Mg 20mM, 0.02% Alcian blue) at 5 days post fertilization (dpf) (120hpf) [95]. Pictures were taken using an Olympus (Antwerp, Belgium) stereomicroscope and camera SZX10 (4x magnification) and Cell B software. Head cartilage and bone skeletons were analyzed using methods previously described [39].

Regarding bone integrity, for the first type of exposure tests we used AB (wildtype) zebrafish. At 10 dpf, 10 zebrafish larvae per treatment were staged, euthanized with an overdose of tricaine (MS-222) and stained ON with 0.05% alizarin red (A5533, CAS 130-22-3, Merck, Overijse, Belgium). On the next morning, fish were rinsed three times with E3 media and observed with an Olympus stereomicroscope and camera SZX10 (4x magnification) and Cell B software. For the second type of experiments, heterozygote parents of the transgenic *Tg (col10a1a:col10a1a-GFP)* line were outcrossed with AB wildtype zebrafish. At 10 dpf, 10 heterozygote transgenic zebrafish larvae per treatment were staged and live stained with alizarin red (dissolved in E3 media at a concentration of 0.1% alizarin plus 500 μL 1M HEPES). Fish was incubated for at least 2 hours, then euthanized with an overdose of tricaine, and mounted in methylcellulose and observed with an epifluorescence stereomicroscope Leica M165 FC (Leica Microsystems, Diegem, Belgium). Pictures were taken, then transferred and analyzed with FIJI (ImageJ2, v. 2.3.0/1.53f).

## RNA extraction and RNAseq

RNA was extracted from pools of 65 larvae at 5 dpf using the RNA mini extraction kit (Qiagen, Hilden, Germany). Details of RNA extraction protocol are described in [22]. Then, the

integrity of total RNA extracts was assessed with BioAnalyzer analysis and provided RIN (RNA integrity number) scores for each sample (Agilent, Santa Clara, CA, USA). cDNA libraries were generated from 100 to 500 ng of extracted total RNA using the Illumina Truseq mRNA stranded kit (Illumina, San Diego, CA, USA) according to the manufacturer's instructions. cDNA libraries were then sequenced on a NovaSeq sequencing system, in $1 \times 100$ bp (single end). Approximatively 20–25 M reads were sequenced per sample. The sequencing reads were processed using Nf-core rnaseq pipeline 3.0 with default parameters and using the zebrafish reference genome (GRCz11) and the annotation set from Ensembl release 103 (www.ensembl.org; accessed 1 May 2020). The analysis for differential gene expression was performed using the DESeq2 pipeline. Pathway and biological function enrichment analysis was performed using the WEB-based "Gene SeT AnaLysis Toolkit" (http://www.webgestalt.org; accessed on 10 November 2022) based on the integrated GO (Gene Ontology), KEGG (Kyoto Encyclopedia of Genes and Genomes), Panther, and WikiPathways databases (all accessed on 10 November 2022 via http://www.webgestalt.org). An additional database was constructed using the Gene-mutant/Phenotype database from zfin (zfin.org; accessed on 6 March 2023). The cut-off values were set for the false discovery rate (FDR) to "adjusted $p$-value $< 0.05$ and the fold change $> 1.5$.

### Chemoinformatics–Cluster analysis of toxprints

We obtained canonical SMILES for each of the 29 compounds within the POP mixture, as well as for Vdr and Rar agonists calcitriol or retinoic acid, respectively, and additional fungicides [96]. Chemical fingerprints (Toxprints) were obtained using Chemotyper [97] (S1 Table), then applying unsupervised classification, to perform a Hierarchical Agglomerative Nesting Clustering with the "Ward's method" [98] using R version 4.2.1 (2022-06-23) — "Funny-Looking Kid" [99], the library "*cluster*" and the function "*agnes*" within that library [100]. The positive controls and list of the analyzed compounds and their fingerprints are in the S1 Table.

### Data and statistical analysis

Generated data were transferred to Prism 9.0.0 (v86), then every data set was tested for normality (*e.g.*, D'Agostino & Pearson test) and equal variances (Bartlett's test). Thus, parametric, or non-parametric tests were performed, each case is indicated in their respective figure. Confidence was assigned at alpha = 95% and a $p$-value of $\leq 0.05$ was considered as significant.

### Supporting information

**S1 Table. List of compounds, SMILES, IUPAC name, concentration and chemical fingerprints within the POP mix and chemicals used for the cluster analysis.** (XLSX)

**S2 Table. Prevalence of craniofacial defects in 5 dpf zebrafish larvae.** Average in percentage and standard deviation values in percentage of the prevalence of micrognathia in zebrafish larvae at 5dpf. Mixed-Effect model (Treatment and Treatment*Phenotype), uncorrected Fisher's LSD test. p values: * <0.05; ** < 0.01; *** <0.001, **** < 0.0001. Asterisk (*) when differences were found against Control, pound sign (#) when differences were found against PFAA alone. (DOCX)

**S3 Table. List of genes differentially expressed upon exposure to POP75× or 125×.** The list is focused on genes for nuclear receptors (Fig 5), genes involved in collagen synthesis (Fig 6), genes for transcription factors, and genes whose expression was found in zebrafish pharyngeal arches at any stage of zebrafish development. The columns give the zebrafish gene name, log

(fold-change) at POP75×, adjusted p-value, log(fold-change) at POP125×, adjusted p-value, and corresponding human gene name.
(XLSX)

**S4 Table. Gene ontology analysis of differentially expressed genes upon POP125× exposure, focused on processes linked to skeletal development.** The table first lists the GSEA analysis using the expression database in zebrafish (highlighted are the genes with decreased expression in the indicated organs = negative enrichment scores), followed by a separate over-representation analysis (ORA) of up (UP)- or down (DOWN)-regulated genes, and finally ORA analysis against the Reactome and GO-molecular fuction (MF) databases.
(XLSX)

## Acknowledgments

The authors would like to thank the GIGA zebrafish platform (H. Pendeville-Samain) for taking care of and delivering the zebrafish larvae, the GIGA imaging platform for their help and support with microscopy, the GIGA genomic platform for sequencing, and the GIGA bioinformatics platform for data analysis.

## Author Contributions

**Conceptualization:** Gustavo Guerrero-Limón, Jérémie Zappia, Marc Muller.

**Data curation:** Gustavo Guerrero-Limón, Marc Muller.

**Formal analysis:** Gustavo Guerrero-Limón, Marc Muller.

**Funding acquisition:** Marc Muller.

**Investigation:** Gustavo Guerrero-Limón, Jérémie Zappia.

**Methodology:** Jérémie Zappia, Marc Muller.

**Software:** Gustavo Guerrero-Limón.

**Supervision:** Jérémie Zappia.

**Validation:** Marc Muller.

**Writing – original draft:** Gustavo Guerrero-Limón.

**Writing – review & editing:** Marc Muller.

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
