## [Decision Letter · Decision Letter 0]

30 Nov 2023

PONE-D-23-36871A realistic mixture of ubiquitous persistent organic pollutants affects bone and cartilage development in zebrafish by interaction with nuclear receptor signaling.PLOS ONE

Dear Dr. Muller,

Thank you for submitting your manuscript to PLOS ONE. After careful consideration, we feel that it has merit but does not fully meet PLOS ONE’s publication criteria as it currently stands. Therefore, we invite you to submit a revised version of the manuscript that addresses the points raised during the review process.

Additional information regarding key experimental details is needed, as pointed out by Reviewer 2. I also recommend that you consider the suggestion by Reviewer 1 to include a mechanistic diagram to explain how chronic POP exposure compromises bone mineralization.

We look forward to receiving your revised manuscript.

Kind regards,

Hans-Joachim Lehmler, PhD

Academic Editor

PLOS ONE

 [This research was funded by the European Union’s Horizon 2020 research and innovation program under the Marie Skłodowska-Curie Innovative Training Network (ITN) program PROTECTED [Grant agreement No. 722634]. G.G-L. was a PROTECTED fellow, M.M. is a "Maître de Recherche" at "Fonds National de Recherche Scientifique (FNRS).].  

[This research was funded by the European Union’s Horizon 2020 research and innovation program under the Marie Skłodowska-Curie Innovative Training Network (ITN) program PROTECTED [Grant agreement No. 722634]. G.G-L. was a PROTECTED fellow, M.M. is a "Maître de Recherche" at "Fonds National de Recherche Scientifique (FNRS). The authors would like to thank the GIGA zebrafish platform (H. Pendeville-Samain) for taking care of and delivering the zebrafish larvae, the GIGA imaging platform for their help and support with microscopy, the GIGA genomic platform for sequencing, and the GIGA bioinformatics platform for data analysis.]

[This research was funded by the European Union’s Horizon 2020 research and innovation program under the Marie Skłodowska-Curie Innovative Training Network (ITN) program PROTECTED [Grant agreement No. 722634]. G.G-L. was a PROTECTED fellow, M.M. is a "Maître de Recherche" at "Fonds National de Recherche Scientifique (FNRS).].

5. We note that Figure(s) 1, 2, 3 and 4 in your submission contain copyrighted images. All PLOS content is published under the Creative Commons Attribution License (CC BY 4.0), which means that the manuscript, images, and Supporting Information files will be freely available online, and any third party is permitted to access, download, copy, distribute, and use these materials in any way, even commercially, with proper attribution. For more information, see our copyright guidelines: http://journals.plos.org/plosone/s/licenses-and-copyright.

a. You may seek permission from the original copyright holder of Figure(s) 1, 2, 3 and 4 to publish the content specifically under the CC BY 4.0 license. 

Reviewers' comments:

Reviewer's Responses to Questions

**Comments to the Author**

1. Is the manuscript technically sound, and do the data support the conclusions?

Reviewer #1: Yes

Reviewer #2: Partly

2. Has the statistical analysis been performed appropriately and rigorously? 

Reviewer #1: Yes

Reviewer #2: Yes

3. Have the authors made all data underlying the findings in their manuscript fully available?

Reviewer #1: Yes

Reviewer #2: Yes

4. Is the manuscript presented in an intelligible fashion and written in standard English?

Reviewer #1: Yes

Reviewer #2: Yes

5. Review Comments to the Author

Reviewer #1: In this research article, Muller and colleagues elaborated the impact of a mixture of ubiquitous persistent organic pollutants (POPs) on bone and cartilage development in zebrafish. This is a complex and specific area of study that involves understanding the effects of environmental contaminants on biological systems. This type of research is crucial for understanding the potential risks and mechanisms of environmental contaminants on developmental processes in organisms, including potential implications for human health. Hence, this research article is relevant for the field.

However, the more informative approaches could be performed such as assessing the expressing of bone related genes (Sox9a, SPP1/OPN, and Col1a1) in addition to the RNA seq (https://doi.org/10.1016/J.SCITOTENV.2021.147989). The authors could compare the results of the reference and highlight their results in Discussion. Besides, did POP induce any oxidative stress, which could be assayed using oxidative stress markers like acridine organe staining, and GSH, SOD, etc metabolic assays. These analysis need to be done to understand the role of POPs in influencing the biological performance of zebrafish.

The authors state in results “Bone mineralization is compromised following 4-day chronic exposure.” This need to be justified with proper clarification citing other references. What lead to such bone demineralization? A mechanistic diagram would be better to explain.

Other Comments

Line 65: Sentence needs to be modified

Line 68: “have addressed” instead of “are addressing”. “have researched” instead of researching. Sentence needs to be modified.

Line 71: “such as this particular” instead of “such as that this particular”

Line 73: “receptor [19], and enhances” instead of “receptor [19], enhances”

Line 81- 86: Segregate the sentence starting from “Some POPs” and ending with “for instance” into two different sentences.

Line 136: “different sub-mixes were tested” instead of “we tested the different sub-mixes”

Line 149: “observed” instead of “photographed”

Line 152: Use technical terms instead of “normal”.

Line 169-171: Sentence starting with “After the different” and ending with “ staining protocol” needs to be modified.

Line 172: “The area of the opercle was measured” instead of “We measured the area”

Line 173 - 175: Sentence starting with “Surviving individuals” and ending with “ including controls” needs to be modified.

Line 187-188: “the transgenic line Tg were used” instead of “we decided to use”

Line 192-193: Sentence starting with “In addition” and ending with “bone structures” needs to be modified.

Line 193: Use technical terms instead of “Looking at”

Line 195: Use technical terms instead of “Surprisingly, looking at”

Line 196 -197: Sentence needs to be modified

Line 198-199: Sentence starting with “Calculating” and ending with “opercle area” needs to be modified and use “The obtained ratio of” instead of “Calculating the ratio between”

Line 199: “Further, this analysis indicates” instead of “Taken together”. “cause” instead of “is causing”

Line 220: “the list of DEGs were reanalysed” instead of “we reanalysed the list of DEGs”

Line 223: “only GSEA identified” instead of “only GSEA also identified”

Line 224 - 225: Sentence starting with “In addition” and ending with “carried out” needs to be modified

Line 227: Rephrase “caught our attention”

Line 232-233: “a network of these zebrafish genes were constructed” instead of “we constructed a network of these zebrafish genes”

Line 236: “It was observed that” instead of “It clearly appears”

Line 253: Rephrase “attracted our attention”

Line 259-260: Rephrase the sentence starting with “Strikingly” and ending with “treatment”

Line 260: Strike out the term “Obviously”

Line 281 - 288: Strike out the term “we” and rephrase the sentences.

Line 307: Strike out the term “our” and specify the list name

Line 314: “focuses” instead of “will focus”

Line 318, 320, 322, & 324: Strike out the term “we” and rephrase the sentences

Line 328: Rephrase “To the best of our knowledge”

Line 329 & 331: Strike out the term “we” and rephrase the sentences

Line 364: “In the experiments carried out mild to severe effects were observed on” instead of “In our experiments, we observed”

Line 372: Strike out the term “looking” and rephrase the sentence

Line 374 -375: Strike out the term “we” and rephrase the sentences

Line 392: Rephrase “We previously showed”

Line 395-396: Strike out the term “we” and rephrase the sentences

Line 405 & 419: Strike out the term “our” and rephrase the sentence

Line 423: Rephrase “our toolbox”

Line 428, 430, & 481: Strike out the term “we” and rephrase the sentences

Line 489-490: Rephrase “no less than”

Line 502: Strike out the term “we” and rephrase the sentences

Some relevant works that can be included and commented in the manuscript are:

Verma, S.K., Nisha, K., Panda, P.K., Patel, P., Kumari, P., Mallick, M.A., Sarkar, B., Das, B., 2020. Green synthesized MgO nanoparticles infer biocompatibility by reducing in vivo molecular nanotoxicity in embryonic zebrafish through arginine interaction elicited apoptosis. Sci. Total Environ. 713, 136521. https://doi.org/10.1016/J.SCITOTENV.2020.136521

Makkar, H., Verma, S.K., Panda, P.K., Jha, E., Das, B., Mukherjee, K., Suar, M., 2018. In Vivo Molecular Toxicity Profile of Dental Bioceramics in Embryonic Zebrafish (Danio rerio). Chem. Res. Toxicol. 31, 914–923. https://doi.org/10.1021/ACS.CHEMRESTOX.8B00129/SUPPL_FILE/TX8B00129_SI_001.ZIP

Reviewer #2: The manuscript by Guerrero-Limon et al describes the toxicity testing of a mixture of 29 POPs at environmentally relevant concentration in zebrafish, with a focus on the bone and cartilage development. The manuscript is well written and the studies are logical and appropriate. There are some comments/clarification needed throughout:

Abstract:

- The first sentence does not add any value

- The number of POPs in the mixture should be stated

Introduction:

- There is not enough context to explain why the effects on bone and cartilage is of concern for POPs. LN 79-81 talks about how there isn’t any studies. So why should we care?

- Some more context on this “realistic mixture” is needed

Methods:

- LN 467: a statement of how the experiment is designed is referenced, but there needs to be more detail of how the solutions were made (LN 461), and experimental chambers, etc. There is barely enough detail to understand there are 7 mixtures created and tested throughout. What concentration of DMSO was used? A blanket statement of “treated for at least 96h” is insufficient. This does not get anywhere close to having enough detail for another lab to reproduce.

- LN 475 – what concentration of MS-222?

Results:

- There needs to be more information on what the 7 mixtures are, rather than just jumping in and saying “POP125+”. At the very least, figure 1 legend should be more descriptive of what the 7 mixtures/submixtures are.

- LN 167 – is 4 day exposure really considered “chronic” if staining happens at 10 dpf. I would state that as continuous exposure. Why the 10 dpf timepoint for bone mineralization?

Discussion:

- This is a well written discussion. The only point that I would have liked to relate why finding structural chemical similarity is a big deal to understanding mixtures?

- Further discussion of how the study for Scandinavian population is going to help NAMs and how does this help with the wildlife population? LN 445 to LN 448 does not make sense.

6. PLOS authors have the option to publish the peer review history of their article (what does this mean?). If published, this will include your full peer review and any attached files.

Reviewer #1: No

Reviewer #2: No

---

## [Author Response · Author response to Decision Letter 0]

8 Jan 2024

checked

 [This research was funded by the European Union’s Horizon 2020 research and innovation program under the Marie Skłodowska-Curie Innovative Training Network (ITN) program PROTECTED [Grant agreement No. 722634]. G.G-L. was a PROTECTED fellow, M.M. is a "Maître de Recherche" at "Fonds National de Recherche Scientifique (FNRS).]. 

done

[This research was funded by the European Union’s Horizon 2020 research and innovation program under the Marie Skłodowska-Curie Innovative Training Network (ITN) program PROTECTED [Grant agreement No. 722634]. G.G-L. was a PROTECTED fellow, M.M. is a "Maître de Recherche" at "Fonds National de Recherche Scientifique (FNRS). The authors would like to thank the GIGA zebrafish platform (H. Pendeville-Samain) for taking care of and delivering the zebrafish larvae, the GIGA imaging platform for their help and support with microscopy, the GIGA genomic platform for sequencing, and the GIGA bioinformatics platform for data analysis.]

We note that you have provided funding information that is not currently declared in your Funding Statement. However, funding information should not appear in the Acknowledgments section or other areas of your manuscript. We will only publish funding information present in the Funding Statement section of the online submission form

[This research was funded by the European Union’s Horizon 2020 research and innovation program under the Marie Skłodowska-Curie Innovative Training Network (ITN) program PROTECTED [Grant agreement No. 722634]. G.G-L. was a PROTECTED fellow, M.M. is a "Maître de Recherche" at "Fonds National de Recherche Scientifique (FNRS).].

5. We note that Figure(s) 1, 2, 3 and 4 in your submission contain copyrighted images. All PLOS content is published under the Creative Commons Attribution License (CC BY 4.0), which means that the manuscript, images, and Supporting Information files will be freely available online, and any third party is permitted to access, download, copy, distribute, and use these materials in any way, even commercially, with proper attribution. For more information, see our copyright guidelines: http://journals.plos.org/plosone/s/licenses-and-copyright.

We would like to emphasize that all the figures presented in this manuscript are originals, that have never been published elsewhere. We actually were in doubt concerning the schematic drawings of alcian blue stained zebrafish larvae in Fig. 1, so we replaced those with a home-made scheme. Please do check that again.

---

## [Decision Letter · Decision Letter 1]

2 Feb 2024

A realistic mixture of ubiquitous persistent organic pollutants affects bone and cartilage development in zebrafish by interaction with nuclear receptor signaling.

PONE-D-23-36871R1

Dear Dr. Muller,

We’re pleased to inform you that your manuscript has been judged scientifically suitable for publication and will be formally accepted for publication once it meets all outstanding technical requirements.

Kind regards,

Hans-Joachim Lehmler, PhD

Academic Editor

PLOS ONE

Additional Editor Comments (optional):

Reviewers' comments:

Reviewer's Responses to Questions

**Comments to the Author**

1. If the authors have adequately addressed your comments raised in a previous round of review and you feel that this manuscript is now acceptable for publication, you may indicate that here to bypass the “Comments to the Author” section, enter your conflict of interest statement in the “Confidential to Editor” section, and submit your "Accept" recommendation.

Reviewer #2: All comments have been addressed

2. Is the manuscript technically sound, and do the data support the conclusions?

Reviewer #2: Yes

3. Has the statistical analysis been performed appropriately and rigorously? 

Reviewer #2: Yes

4. Have the authors made all data underlying the findings in their manuscript fully available?

Reviewer #2: Yes

5. Is the manuscript presented in an intelligible fashion and written in standard English?

Reviewer #2: Yes

6. Review Comments to the Author

Reviewer #2: The authors have addressed all my concerns and I support the acceptance of this paper. Please make sure the data is shared with per the data policy

7. PLOS authors have the option to publish the peer review history of their article (what does this mean?). If published, this will include your full peer review and any attached files.

Reviewer #2: No

---

## [Editor Report · Acceptance letter]

19 Mar 2024

PONE-D-23-36871R1 

PLOS ONE

Dear Dr. Muller, 

I'm pleased to inform you that your manuscript has been deemed suitable for publication in PLOS ONE. Congratulations! Your manuscript is now being handed over to our production team.

Kind regards, 

on behalf of

Dr. Hans-Joachim Lehmler 

Academic Editor

PLOS ONE